



# Ice nucleation on surrogates of boreal forest SOA particles: effect of water content and oxidative age

Ana A. Piedehierro[1], André Welti[1], Angela Buchholtz[2], Kimmo Korhonen[2], Iida Pullinen[2], Ilkka Summanen[2], Annele Virtanen[2], and Ari Laaksonen[1,2]

[1]Finnish Meteorological Institute, Helsinki, Finland.
[2]Department of Applied Physics, University of Eastern Finland, Kuopio, Finland.

*Correspondence to:* Ana A. Piedehierro (ana.alvarez.piedehierro@fmi.fi)

**Abstract.** $\alpha$-pinene is an abundant volatile organic compound (VOC) emitted by boreal forests and a source of atmospheric Secondary Organic Aerosol (SOA). This precursor is commonly used as a model compound for SOA studies representing boreal forest emissions. $\alpha$-pinene SOA particles can have a highly viscous solid or semi-solid phase state depending on water content, temperature and oxidative age. The phase state (or viscosity) of SOA particles has multiple effects on the chemical

and physical processes in which SOA particles are involved; one of the affected processes is ice formation.

We investigate the effect of water content and oxidative age on ice nucleation using $100\,\text{nm}$ quasi-monodisperse particles of boreal forest SOA surrogates . Ice nucleation experiments are conducted in the temperature range between 210 and $240\,\text{K}$ and from ice to water saturation using the Spectrometer for Ice Nuclei (SPIN). The effect of the particle water content on the ice nucleation process is tested by preconditioning $\alpha$-pinene SOA at different humidity (40%, 10% and <1% $\text{RH}_\text{W}$). The influence

of the particle oxidative age is tested by varying their O:C ratio (oxygen-to-carbon ratios, O:C $\sim$ 0.45, 0.8, 1.1). To assess the suitability of $\alpha$-pinene as a model compound to study the ice nucleation properties of boreal forest SOA and to confirm the atmospheric relevance of our findings, we compare them to measurements of SOA using pine-needle oil or Scots pine tree emissions as precursors.

The ice nucleation measurements show that surrogates of boreal forest SOA particles promote only homogeneous ice formation.

An effect of preconditioning humidity on homogeneous ice nucleation could be observed. Contrary to the expected behavior, homogeneous freezing is suppressed for SOA particles with high water content (preconditioned at 40% $\text{RH}_\text{W}$) and was only observed for SOA preconditioned at low $\text{RH}_\text{W}$ ($\leq 10\%$). No dependence of homogeneous freezing on the SOA oxidative age was observed. The results can be explained by a significant change of particulate water diffusivity as a function of humidity (from 10% to 40% $\text{RH}_\text{W}$) at $293\,\text{K}$, where the aerosol is preconditioned. On dry SOA particles, water diffusion into the

particle is slow enough to form a core-shell morphology with an outer layer that can equilibrate within the timescale of the experiment and freeze homogeneously. On SOA particles with higher water content, water diffuses faster into the particle, delaying equilibration at the particle surface and preventing the formation of a diluted shell, which can delay homogeneous freezing. To predict if a core-shell develops, we propose that the partial water vapor pressure particles are exposed to prior to an experiment can serve as an indicator.





# 1 Introduction

Heterogeneous ice formation by ice nucleating particles (INP) allows the formation of cirrus clouds at lower humidity than required for ice formation by homogeneous freezing of solution droplets, which is determined by the water activity criterion (Koop et al., 2000). Typical cirrus cloud INPs are water insoluble particles such as mineral dusts, fly ash, metallic particles (DeMott et al., 2003) or soots (Bond et al., 2013). Prompted by the realization that secondary organic aerosol (SOA) particles can exist in a highly viscous, (semi-) solid state (Zobrist et al., 2008; Virtanen et al., 2010), the possibility that SOA particles could act as INPs has been investigated in the recent years (Wang et al., 2012; Schill et al., 2014; Ignatius et al., 2016; Ladino et al., 2014; Möhler et al., 2008; Prenni et al., 2009; Charnawskas et al., 2017; Wagner et al., 2017).

SOA particles are composed of oxidation products of volatile organic compounds (VOC), some of which are water soluble. In contrast to low viscosity (more liquid-like) particles, highly viscous aerosol are slow to take up or lose water or other vapors in response to variations in gas-phase composition (Mikhailov et al., 2009; Koop et al., 2011; Shiraiwa et al., 2013; Yli-Juuti et al., 2017). This resistance is particularly pronounced under dry conditions or low temperatures where SOA particles can exist in a highly viscous, or glassy state (Koop et al., 2011; Virtanen et al., 2010; Zobrist et al., 2008).

Upon updraft-driven humidification in the atmosphere, (semi-) solid amorphous organic aerosol could act as INP in several ice nucleation mechanisms. SOA particles that remain glassy during humidification could trigger ice formation via deposition nucleation (Murray et al., 2010). Upon humidification beyond the amorphous deliquescence relative humidity, water diffusion might be slow enough to form core-shell morphologies in particles with a highly viscous glassy matrix, allowing ice formation via immersion freezing (Berkemeier et al., 2014; Lienhard et al., 2015). This situation could take place during fast updrafts (> $3\,\mathrm{m\,s^{-1}}$) in which the equilibration time with the surrounding humidity is limited (Lienhard et al., 2015; Price et al., 2015). Typical equilibration times for $100\,\mathrm{nm}$ $\alpha$-pinene SOA particles range from hours at $220\,\mathrm{K}$ to minutes at $230\,\mathrm{K}$ (Price et al., 2015). Continued humidification of the organic particles leads to complete liquefaction and homogeneous freezing as the only possible ice formation mechanism (Koop et al., 2011).

Previous studies found that the ice nucleation (IN) ability of SOA particles varies between species: naphthalene-derived (Wang et al., 2012), and methylglyoxal with methylamine (Schill et al., 2014) particles were classified as effective, heterogeneous INP, exhibiting ice formation onsets at humidities clearly below homogeneous freezing conditions. $\alpha$-pinene SOA was indicated to be ineffective at nucleating ice heterogeneously (Möhler et al., 2008; Ladino et al., 2014; Prenni et al., 2009; Charnawskas et al., 2017; Wagner et al., 2017) while Ignatius et al. (2016) found $\alpha$-pinene SOA particles to be effective INPs in the deposition mode. Although the rapid cooling of the aerosol during most experiments could result in the development of a core-shell morphology, none of the above studies observed ice formation by immersion freezing. Using a cloud particle model capable of simulating water diffusion in individual aerosol particles, Fowler et al. (2020) have recently shown that at temperatures between 200 and $220\,\mathrm{K}$ a water layer condenses on $\alpha$-pinene SOA particles, which can freeze homogeneously. In accordance with laboratory studies showing only homogeneous ice formation of $\alpha$-pinene SOA particles, the same model indicates that at higher temperatures, water diffuses into particles, and freezing only takes place at humidities high enough to enable the freezing of solution droplets. Mie resonance measurements in levitating, highly viscous aerosol particles show

steep gradients in water content between the core and outer layer (Bastelberger et al., 2018) supporting the interpretation that viscous SOA can adopt a core-shell structure, where the outer layer can freeze homogeneously. We present experimental results of ice formation of atmospherically relevant surrogates of SOA particles, generated from biogenic emissions. To test the effect of water content and core-shell formation, $\alpha$-pinene SOA particles with different diffusivity are generated by varying the

oxidative ageing and exposure of particles to different levels of humidity prior to cooling. The atmospheric relevance of the ice nucleation results from the $\alpha$-pinene SOA is confirmed by comparing to experiments with SOA particles formed from natural, boreal forest VOC.

## 2    Methods

The SOA Ice Nucleation Experiment (SINE) campaign was carried out at the University of Eastern Finland Aerosol Physics

Laboratory in June - July 2019. The experiments focused on studying the IN efficiency of SOA formed from boreal forest emissions and the influence of particle water content and oxidation level on IN activity.

### 2.1    General experimental setup

A schematic of the experimental setup is depicted in Figure 1. SOA was generated either in an atmospheric simulation chamber (ASC; Faiola et al. (2019); batch chamber) or a potential aerosol mass reactor (PAM; Aerodyne Research Inc.; Kang et al.

(2007), Lambe et al. (2011); oxidative flow reactor). The details of the two SOA formation procedures are described in sections 2.2.1 and 2.2.2.

The SOA precursor concentrations (VOC, Tab. 1) were monitored by a high-resolution time-of-flight proton transfer reaction mass spectrometer (PTR-MS, Ionicon model 8000). The size distribution and the oxidative age (O:C ratio, Tab. 1) of the SOA particles were monitored using a scanning mobility particle sizer (SMPS: TSI, Inc. model DMA 3082, CPC 3775) and a high-

resolution time-of-flight aerosol mass spectrometer (HR-AMS: Aerodyne Research Inc., DeCarlo et al. (2006)), respectively. For the ice nucleation measurements with a spectrometer for ice nuclei (SPIN; Droplet Measurement Technologies) the aerosol was size-selected using a differential mobility analyzer (DMA; TSI, model 3082) and preconditioned at 40%, 10% or <1% $RH_W$.

### 2.2    Aerosol generation

#### 2.2.1    $\alpha$-pinene SOA with different O:C generated with the PAM reactor

We used a PAM reactor to form $\alpha$-pinene SOA particles of different oxidative ages, characterized by their O:C ratio. A syringe pump (Nexus 3000, Chemyx Inc.) was used to create a constant injection of $\alpha$-pinene into a nitrogen carrier gas flow heated to 60 °C, which was introduced into the PAM reactor. In the PAM reactor, particles are formed by a) dry ozonolysis, b) wet ozonolysis, or c) wet photooxidation with OH radicals. The $O_3$ concentration, and in the photooxidation experiments the

irradiation level, were adjusted to create SOA particles with a specific O:C ratio (see Tab. 1 for settings). Low-O:C SOA





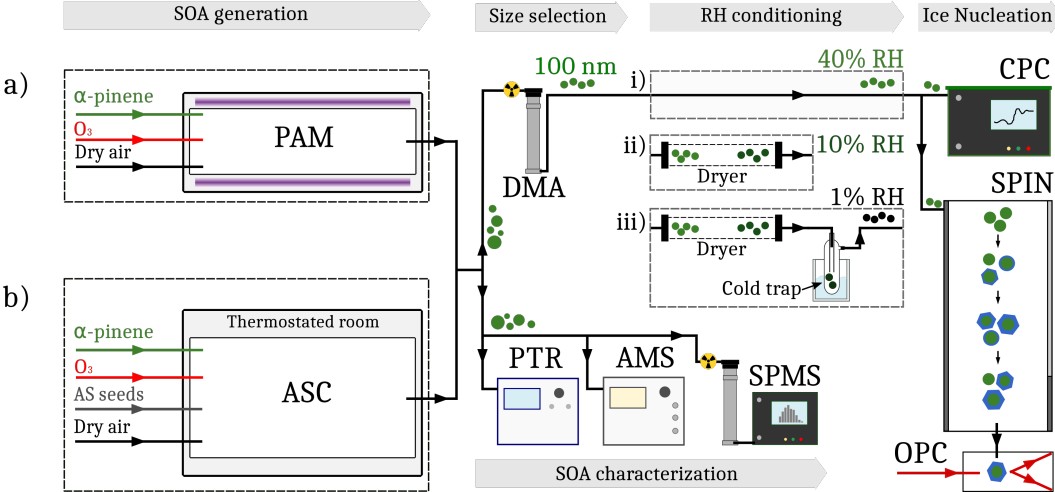

**Figure 1.** Experimental set-up during SINE. The set-up consists of five stages: SOA generation, size selection, SOA characterization, RH conditioning, and ice nucleation measurement. Two methods of SOA generation were used: a) homogeneous nucleation inside the PAM reactor and b) seeded nucleation inside the ASC. For the PAM experiments, three configurations of the RH conditioning stage were used before measuring the IN activity of the SOA particles: i) 40%, ii) 10% or iii) <1% $RH_W$).

(O:C ~0.45) was produced in wet and dry ozonolysis experiments (#1 - #7). The photooxidation experiments (#8 - #12) generated medium- (O:C ~0.9) and high-O:C (O:C ~1.1) SOA. For dry ozonolysis, dry nitrogen and oxygen were used as carrier gases resulting in $RH_W$ of <2% in the sample flow exiting the PAM reactor. For wet ozonolysis and photooxidation experiments, the nitrogen flow was humidified to create $RH_W$ between 40% and 50% inside the reactor. The OH radicals in the

photooxidation experiments were created from the photolysis of $O_3$ at 254 nm and consecutive reaction of the formed $O(^1D)$ with water vapor. The composition of the SOA particles formed in the PAM reactor were held constant over the course of the 4-5 hours needed for the IN measurements. Between two experiments, the PAM reactor and all connected tubes were flushed with nitrogen or purified air for several hours. Before a new experiment started, the PAM reactor ran for at least 30 min at similar $O_3$ concentrations and irradiation as during the experiment but without VOC to ensure low levels of background SOA

(< 1 $\mu g\,m^{-3}$).

### 2.2.2 Boreal forest surrogate SOA from ASC

A second set of experiments was conducted utilizing the 12 $m^3$ collapsible PTFE ASC to form SOA under atmospheric conditions (see Fig. 1). Aqueous ammonium sulfate particles were generated with an atomizer (Aerosol Generator Model 3076, TSI), subsequently dried below their efflorescence point in a silica gel dryer and injected into the ASC to reach seed concen-

trations of 5.0 – 7.5e4 $cm^{-3}$ at the start of VOC ozonolysis. For experiments #13 and #15, $\alpha$-pinene or pine needle oil were added to the chamber from diffusion sources. The injection times were adjusted to reach the desired 5 - 50 ppb in the ASC. The emissions from six 10 year-old pine saplings had been collected onto stainless steel multibed adsorbent cartridges containing





**Table 1.** Range of SOA formation and preconditioning parameters for each PAM reactor and ASC experiment. The VOC concentrations [VOC] reported for PAM experiments correspond to average concentrations while for ASC refer to starting concentrations. The variability of parameters during individual experiments were $RH_W$ ±1%, [VOC] ±7.0 ppb and O:C ratio ±0.03.

| Exp # | Reactor | VOC | [VOC] | Formation RH$_W$ | Dominant oxidant | O:C ratio | Seed | Preconditioning RH$_W$ | Fig. |
| | | | [ppb] | [%] | | | | [%] | |
|---|---|---|---|---|---|---|---|---|---|
| 1 | PAM | $\alpha$-pinene | 99 | 47 | O$_3$ | 0.46 | - | 40 | 2.a |
| 2 | PAM | $\alpha$-pinene | 131 | 40 | O$_3$ | 0.45 | - | <10 | 2.b |
| 3 | PAM | $\alpha$-pinene | 127 | 50 | O$_3$ | 0.44 | - | <10 | 2.b |
| 4 | PAM | $\alpha$-pinene | 137 | 50 | O$_3$ | 0.45 | - | <1 | 2.c |
| 5 | PAM | $\alpha$-pinene | 129 | 40 | O$_3$ | 0.45 | - | <1 | 2.c |
| 6 | PAM | $\alpha$-pinene | 122 | 2 | O$_3$ | 0.45 | - | <2 | 2.c |
| 7 | PAM | $\alpha$-pinene | 90 | 1 | O$_3$ | 0.46 | - | <1 | 2.c |
| 8 | PAM | $\alpha$-pinene | 105 | 44 | OH | 0.91 | - | 40 | 2.d |
| 9 | PAM | $\alpha$-pinene | 95 | 43 | OH | 0.88 | - | <10 | 2.e |
| 10 | PAM | $\alpha$-pinene | 98 | 44 | OH | 0.86 | - | <1 | 2.f |
| 11 | PAM | $\alpha$-pinene | 120 | 40 | OH | 1.05 | - | 40 | 2.g |
| 12 | PAM | $\alpha$-pinene | 107 | 42 | OH | 1.16 | - | <10 | 2.h |
| 13 | ASC | pine-needle-oil | 21 * | <15 | O$_3$ | 0.6 | (NH$_4$)$_2$SO$_4$ | <10 | 5.a |
| 14 | ASC | pine emissions | 6 * | <15 | O$_3$ | 0.53 | (NH$_4$)$_2$SO$_4$ | <10 | 5.b |
| 15 | ASC | $\alpha$-pinene | 6 * | <15 | O$_3$ | 0.61 | (NH$_4$)$_2$SO$_4$ | <10 | 5.c |

\* concentration of VOCs detectable with PTR-MS

Tenax TA and Carbograph adsorbent (Markes International, Inc.) during the springtime of 2019 (similar procedure as described by Faiola et al. 2019). For experiment #14, 4 of these tubes (equivalent to 30 h of emissions) were introduced into the ASC by flushing the cartridges with nitrogen for 20 min, while they were heated to 473 K. Mixing in the ASC was achieved by an additional flow of purified compressed air (> 50 L min$^{-1}$). O$_3$ was added to the chamber from a custom-built ozone generator, immediately afterwards, to start the ozonolysis reactions.

5   The precursor concentration in the ASC was monitored with the PTR-MS. The particle size and composition were monitored continuously with the SMPS and the HR-AMS. The IN measurements started after the maximum in the SOA mass concentration was reached, ensuring a constant particle composition during the IN measurements. Between experiments, the ASC was flushed for several hours with purified compressed air. Additionally, chemical cleaning was conducted by filling the ASC with

10   high concentrations of O$_3$ (>500 ppb) for 30 – 60 min. These chemical cleaning intervals were followed by at least 10 h of flushing.





## 2.3 Ice nucleation measurements

The ice nucleation ability of $100\,nm$ SOA particles was measured with the SPIN instrument (Garimella et al., 2016). SPIN is a parallel plate, continuous flow diffusion chamber (CFDC). The specific instrument (SPIN5) used in this study has been modified to perform low temperature experiments (Welti et al., 2020). Inside the SPIN chamber, particles are exposed to RH- and T-conditions (temperature range 210 - 240 K and from ice to water saturation) relevant for ice formation in cirrus clouds. Ice formation during a residence time of $10\,s$ is detected with an optical particle counter (OPC). The ratio of ice forming particles, measured with the OPC and the number of particles introduced into SPIN, measured with a condensation particle counter (CPC; Airmodus A20), is reported as activated fraction (AF). The evaporation section at the end of the chamber guarantees that droplets are not counted as ice with the OPC. In this study, the size-selected SOA particles were exposed to different humidity conditions before entering the SPIN, to probe the effects of particle water content on IN activity. The samples were introduced a) directly into the SPIN, b) through a silica gel diffusion dryer, or c) sequentially through a silica gel diffusion dryer and a liquid nitrogen cold trap (see Fig. 1). The preconditioning led to sample $RH_W$ at the SPIN inlet (Vaisala HMP110 humidity sensor) of 40% , 10%, or <1%, respectively. Later, we refer to these precondition settings as wet, dry and super dry, respectively.

## 3 Results and Discussion

### 3.1 Effect of O:C ratio and water content on ice formation conditions

The effects of oxidative age (approximated by the O:C ratio) and water content of $\alpha$-pinene SOA particles on ice formation were investigated. Note, that the water content of the particles was preconditioned after generation and before measuring their ice nucleation ability with the SPIN. The AF-spectra of $\alpha$-pinene SOA particles generated with different O:C and preconditioned at different humidity are shown in Fig. 2.

The O:C ratio appears to have a minor impact on the conditions of ice formation of the $\alpha$-pinene SOA particles. Schill and Tolbert (2012) reported that the O:C ratio of organic acids affects heterogeneous ice nucleation and suggested that this could also be the case for $\alpha$-pinene SOA. Aged particles (higher O:C ratio) would nucleate ice at lower humidity due to higher surface hydrophilicity, allowing the adsorption of an ice-like layer from which the ice phase can develop. The decrease in ice formation onset humidity was strongest for 0.3 < O:C < 0.5. In the range of O:C ratios covered here (0.45-1.1) no pronounced change in ice formation humidity was observed. Fig. 3 summarizes humidity conditions where AF = 1% were reached in dependence of O:C ratios. The humidity needed to reach 1% AF is close to humidities at which ammonium sulfate solution droplets freeze homogeneously (e.g., reported in Welti et al. (2020)). This suggests that homogeneous freezing dominated the ice formation of SOA particles with O:C ratios between 0.45 and 1.1, regardless of the oxidative age itself.

A strong dependence of ice formation on SOA water content was observed. Dry and super dry SOA particles ($RH_W = 10\%$ and <1%, middle and right column in Fig. 2) form ice at humidities close to the homogeneous freezing line of solutes, the so-called Koop line (Koop et al., 2000), suggesting homogeneous freezing. Wet preconditioning of the SOA (at 40% $RH_W$, left





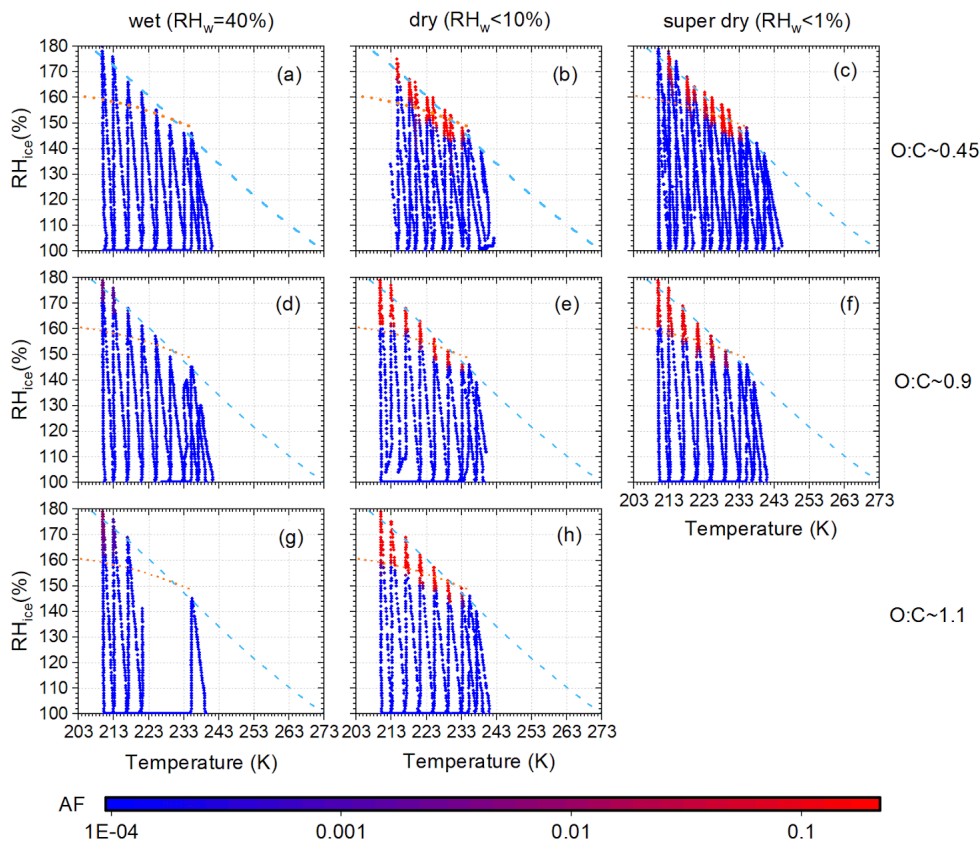

**Figure 2.** AF measurements for $\alpha$-pinene O:C $\sim$0.45, 0.9 and 1.1 (top, middle and bottom row) and 4%, 10%, <1% $RH_W$ (left, middle and right column). The dashed, light blue line marks water saturation. Conditions of homogeneous freezing (AF=1%) of $\alpha$-pinene SOA particles are shown as orange, dotted lines. Homogeneous freezing conditions were calculated following Appendix B in Welti et al. (2020) and using the growth factor for $\alpha$-pinene SOA given in Varutbangkul et al. (2006).

column in Fig. 2) almost completely suppressed the formation of ice up to water saturation. SOA particles preconditioned at higher $RH_W$ contain more water, which acts as a plasticizer, and they are therefore less viscous. It was expected that the higher water content of the particles would facilitate the diffusion of condensed water molecules, enabling the homogeneous freezing of SOA solution droplets. To our knowledge, a retardation of homogeneous freezing of SOA particles with higher water content has not been observed experimentally and only recently been suggested by Fowler et al. (2020), based on simulations. Fig. 4.
5   illustrates the mechanism how water content could impact homogeneous freezing of SOA particles during SPIN measurements.

When dry SOA particles (Fig. 4a) are exposed to the elevated humidity inside the SPIN, water diffusion towards the particle core is slow enough to create a core-shell structure with a steep gradient in water content. The liquid outer layer equilibrates





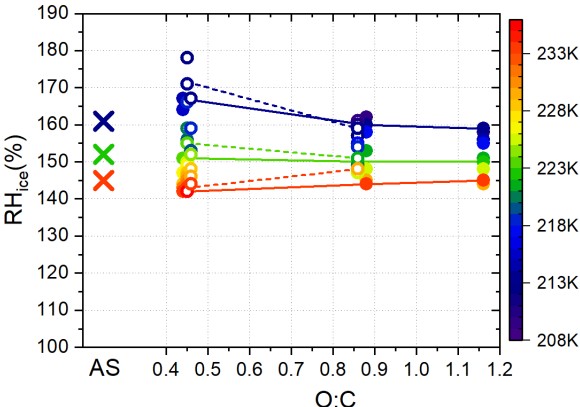

**Figure 3.** Relative humidity of ice formation on 1% of 100 nm $\alpha$-pinene SOA particles as function of O:C ratio (circles) in comparison to ammonium sulfate (AS, shown as crosses). AS data from Welti et al. (2020). Open circles indicate experiments with $\alpha$-pinene particles preconditioned at $RH_W < 1\%$, filled circles $RH_W = 10\%$. Trendlines are shown for experiments at 213 K, 223 K and 233 K. Temperature is indicated by the color scale.

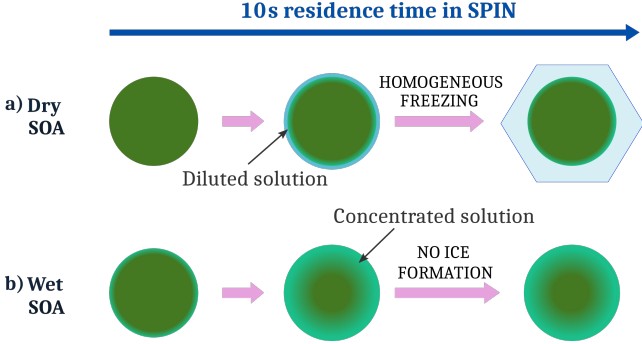

**Figure 4.** Suggested freezing mechanism for SOA particles with different water content.

to the SPIN humidity and freezes when homogeneous freezing conditions are reached. This mechanism is similar to the one suggested by Fowler et al. (2020).

For SOA particles with higher water content (Fig. 4b), water diffuses faster into the particle, leading to a core-shell morphology with a more concentrated outer layer. The outer shell of the particle does not reach equilibrium and inhibits homogeneous freezing. The mechanism depicted in Fig. 4 suggests that slower diffusion enables the outer shell to equilibrate faster with the gas phase. For a constant amount of water uptake (within 10 s residence time in SPIN), wet SOA particles incorporate water faster into the particle, resulting in a more concentrated droplet or shell. With longer residence times equilibrium could be reached and homogeneous freezing would theoretically be observed again.





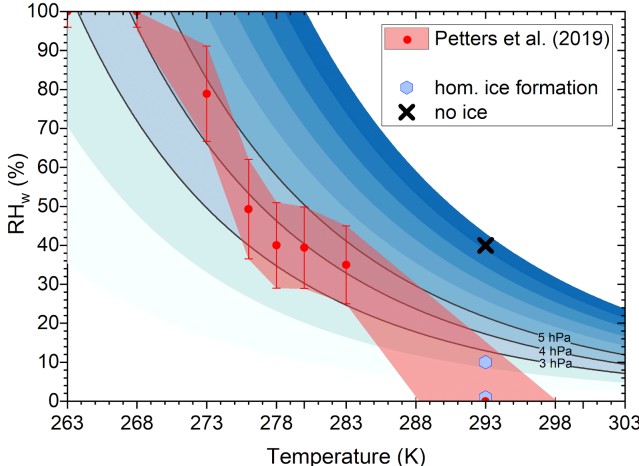

**Figure 5.** Comparison of SOA pre-treatment conditions to SOA phase transition (viscosity 4e5 to 6e6, red envelope) from Petters et al. (2019). We note that the phase transition coincides with a partial water vapor pressure (shown in blue) of 3-5 hPa. Homogeneous ice nucleation was only observed for SOA particles exposed to low water vapor pressure during pre-conditioning.

Comparing the preconditioning humidities to the phase diagram of RH-induced glass transition as a function of temperature for $\alpha$-pinene SOA in Fig. 5 (reproduced from Petters et al. (2019)) suggests that SOA particles were in a liquid-like state before entering the SPIN instrument for the wet preconditioning (cross in Fig. 5, indicating suppression of ice formation). Dry preconditioned particles, for which homogeneous ice formation was observed, were likely glassy (blue hexagons in Fig.

5). The growth factor of $\alpha$-pinene SOA at 40% $RH_W$, measured by Varutbangkul et al. (2006), is 1.00675, which would correspond to 13% mole fraction of water in the SOA particles assuming an average molecular weight of $200\,\mathrm{g\,mol^{-1}}$ and density of $1.5\,\mathrm{g\,cm^{-3}}$. This indicates that at this humidity SOA particles can take up a substantial amount of water molecules. Evaporation studies on $\alpha$-pinene SOA (Li et al., 2019) also indicate that at room temperature and 40% $RH_W$ $\alpha$-pinene SOA particles present negligible kinetic limitations, evaporating in a nearly liquid-like state, while reduced $RH_W$ conditions (<1%

$RH_W$) lead to slower evaporation rates, due to the limited bulk diffusivity of the particles. According to Fig. 5, once the SOA particles enter the SPIN instrument, the particles solidify abruptly with the decreasing temperature, but seem to retain a preconditioning humidity dependent water diffusion rate, governing the equilibration timescale.

We propose that a partial water vapor pressure of 3-5 hPa can be used as marker condition to distinguish water diffusion rates of $\alpha$-pinene particles. Slower water diffusion into particles kept below 3-5 hPa leading to faster equilibration and homo-

geneous ice formation, while particles crossing this condition show faster diffusion and require longer equilibration times. To better understand the equilibration timescales of the wet or (super-) dry preconditioned particles and consequently the time dependence of homogeneous ice formation on SOA particles, additional experiments are needed. It would be beneficial to investigate in detail the correlation between the homogeneous ice nucleation ability of SOA and water content by extending the T-, and RH- conditions at which SOA are preconditioned to cover the threshold conditions shown in Fig. 5.




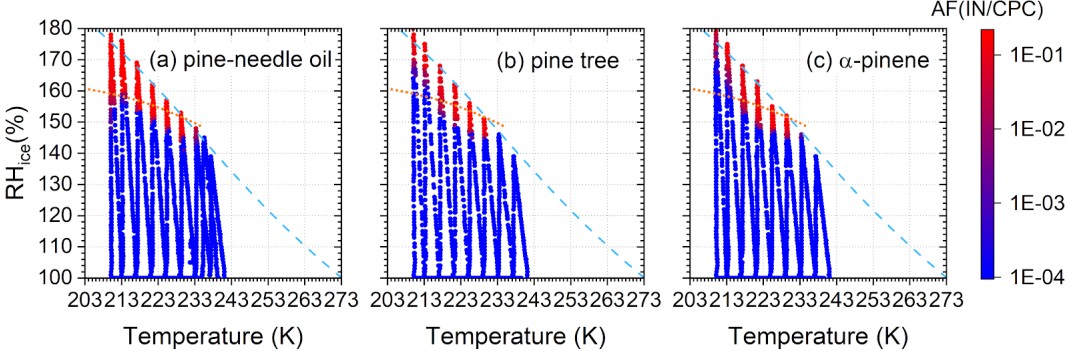

**Figure 6.** Activated fraction of $100\,\mathrm{nm}$ SOA particles. The precursor type is indicated in each subfigure. The light blue, dashed line marks water saturation. Conditions where 1% soluble particles freeze homogeneously according to the parameterization of Koop et al. (2000) are shown as orange curves, dotted lines.

## 3.2 $\alpha$-pinene as proxy for boreal forest emissions

After isoprene, monoterpenes are the second most emitted volatiles by plants (Sindelarova et al., 2014). Among monoterpenes, $\alpha$-pinene is the most abundant VOC emitted from the boreal forest (Wang et al., 2018). We studied the ice nucleation conditions of SOA particles nucleated from Scots pine tree emissions and compared them to two precursor proxies: pine-needle oil (Sigma Aldrich) and $\alpha$-pinene. The SOA particles were generated in an ASC (Fig.1) and their IN ability tested between $210\,\mathrm{K}$ and $240\,\mathrm{K}$ and from ice to water saturation (Fig. 6). This is the first study reporting the IN ability of SOA from real pine emissions representative of boreal forest environments.

The particles nucleated ice only at humidities close to the Koop line between approx. $215\,\mathrm{K}$ and $235\,\mathrm{K}$, showing a steep onset when humidity increased during the RH-scans. Compared to the Koop parameterization, ice formation on Scots pine tree (Fig. 6b) and $\alpha$-pinene (Fig. 6c) SOA is retarded to higher supersaturations in the lowest range of temperatures, similar to dry conditioned $\alpha$-pinene SOA generated in the PAM reactor. In summary, the three tested precursors showed similar ice nucleation conditions, typical for homogeneous freezing, indicating that $\alpha$-pinene is an appropriate proxy for IN by SOA from boreal forests emissions.

## 3.3 Comparison to previous measurements of $\alpha$-pinene SOA ice formation conditions

The 1% AF values extracted from the RH- scans with dry and super dry $\alpha$-pinene SOA are summarized in Fig. 7 together with data from the literature.

Conditions of 1% AF ice formation on $\alpha$-pinene SOA, measured in this study agree well with previous measurements reported by Möhler et al. (2008), Ladino et al. (2014), Wagner et al. (2017), and Charnawskas et al. (2017). In these studies, SOA particles were generated at T-, and RH-conditions comparable to the experiments for dry preconditioned SOA presented here. These studies concluded that $\alpha$-pinene SOA is a poor ice nucleating particle for heterogeneous ice formation or that they





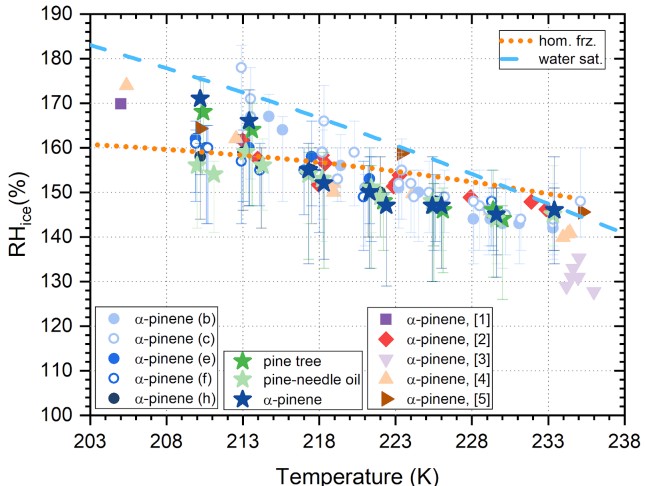

**Figure 7.** Conditions of ice formation on 1% of 100 nm $\alpha$-pinene SOA particles. Circles and stars represent the results from the PAM and ASC experiments, respectively. Open and filled symbols mark $RH_W$=1% and 10% preconditioning, respectively. $\alpha$-pinene SOA ice nucleation data from previous studies are shown for comparison: [1] Möhler et al. (2008), [2] Ladino et al. (2014), [3] Ignatius et al. (2016), [4] Wagner et al. (2017), [5] Charnawskas et al. (2017). The light blue dashed line marks water saturation. Conditions where 1% soluble particles freeze homogeneously according to the parameterization of Koop et al. (2000) are shown as dotted, orange curve.

exclusively nucleate ice homogeneously. However, the data deviates from the Koop line. Humidity conditions of ice formation decrease monotonically with increasing temperature, approximately parallel to the water saturation line, i.e. showing earlier onsets for the warmest temperatures and delayed onset at the cold temperatures. Ignatius et al. (2016) argued that ice forming below the Koop line should be classified as heterogeneous ice formation. However, homogeneous ice formation below the Koop

line was also observed during reference measurements using ammonium sulfate solution droplets with the SPIN instrument (Welti et al., 2020). A steep increase in the AF at a certain humidity during RH-scans is an additional indication that the driving ice formation mechanism was homogeneous freezing, despite the lower humidities for ice nucleation in comparison to the Koop parametrization.

Delayed onsets of homogeneous freezing above the humidity defined by the water activity criterion (Koop et al., 2000) at

10 low temperatures could indicate that particles are not in equilibrium before freezing in SPIN. A similar offset for homogeneous freezing of sulfuric acid particles at low temperature has been observed in the AIDA chamber (Möhler et al., 2003), which was also attributed to diffusion-limited uptake of water vapor during cooling. The systematic delay of homogeneous freezing towards higher humidity, suggests a decreasing, low water diffusion rate of SOA towards low temperature. Fowler et al. (2020) pointed out the same aspect, and proposed even an inhibition of homogeneous ice nucleation below 200 K. Experimental

evidence of homogeneous freezing inhibition at such low temperatures had been previously reported by Murray (2008) for citric acid solution droplets.

## 4    Conclusions

We have investigated the influence of particle water content and oxidative age on the ice nucleation ability of boreal forest SOA surrogates, using $\alpha$-pinene SOA generated in a PAM reactor as a model compound. The suitability of $\alpha$-pinene SOA as a proxy for the IN ability of more complex boreal forest surrogates was addressed by comparing IN measurements of SOA

from two other precursors (pine-needle oil and Scots pine tree VOC) generated in an ASC under more atmospherically relevant conditions. For the first time, the IN ability of pure lab-generated SOA from real pine emission has been measured.

Boreal forest SOA surrogates from all precursors were found to be inefficient ice nucleating particles, in agreement with previous studies (Möhler et al., 2008; Ladino et al., 2014; Wagner et al., 2017; Charnawskas et al., 2017). Our observations indicate that the IN ability of $\alpha$-pinene SOA can be considered representative of more complex SOA produced in monoterpene-

dominated precursor mixes from boreal environments.

Homogeneous ice nucleation was observed for $\alpha$-pinene SOA preconditioned at low $RH_W$ ($\leq 10\%$) and contrary to the expected behavior, homogeneous freezing was suppressed for SOA with higher water content (40% $RH_W$). The onset humidity for homogeneous freezing did not depend on the oxidative age. The experiments indicate that SOA water content was the main variable controlling the onset humidity for homogeneous freezing at a certain temperature and point to a dependence on the

water diffusion rate.

$\alpha$-pinene SOA with low water content, presumably in a highly viscous state, did not act as INP, but a decreased water diffusion rate into the particle allowed the formation of a core-shell morphology, enabling the homogeneous freezing of the diluted outer layer. In contrast, SOA with higher water content, into which water diffused more efficiently during the residence time in SPIN, completely liquefied or developed a core-shell morphology in which the liquid phase was highly concentrated,

inhibiting homogeneous freezing. The coincidence of SOA phase transition conditions with partial water vapor pressures between 3-5 hPa also point at the potential dominant role of water content in particle properties like viscosity or diffusivity that affected the SOA ice nucleation behavior as measured by SPIN.

Further investigations should include SOA generation under a wider range of RH and T conditions and measurements of initial viscosity for a clearer connection between the ice nucleation measurements and the freezing pathways leading to those

observations.

*Data availability.* Data sets are available from the authors upon request.

*Author contributions.* AV and AL proposed the initial research question. AAP and AW conducted the ice nucleation experiments with contributions from KK. AB, IP and IS were in charge of particle formation, characterization, and the corresponding data analysis. AAP and AW prepared the manuscript with contributions from AB, AL, and AV. All authors commented the manuscript. AV and AL acquired funding

and supervised the project.





*Competing interests.* The authors declare that they have no conflict of interest.

*Acknowledgements.* We thank Zijun Li (Rex) and Pasi Miettinen for their support during the campaign. This work was supported by the Academy of Finland, C-Main project (grant no. 309141), and the Center of Excellence program (grant no. 307331).



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
