# Peer review of "Ice nucleation on surrogates of boreal forest SOA particles: effect of water content and oxidative age"

_Atmospheric Chemistry and Physics, 2021_

## Referee Comment (RC1)

**Review of "Ice nucleation on surrogates of boreal forest SOA particles: effect of water content and oxidative age"**

I have general comments on the abstract below. Generally, I think the paper is well written and a nice study that adds to the body literature. I also have a comment on figure 2, which is fairly trivial.

- Abstract is a little longer than it needs to be. I feel like the first paragraph could be removed to add a bit more focus.
- Was the core-shell formation observed in the measurements – could this be said explicitly in the abstract?
- You clearly show the results of preconditioning the SOA at different humidities in figure 2. This is a key figure for the paper and well-presented. For those not well versed with the CFDCs I think it is necessary to explain what the steps in the plots are (scans). There is also a typo in the legend, which refers to 4% RH instead of 40%.

That said, upon detailed reading I am confused about how it works.

The observations seem clear:

- preconditioning the aerosol at 40% RH allows the aerosol to take on more water at room temperature. When you then transfer the aerosol to the CFDC these aerosol particles nucleate ice close to the threshold for homogeneous nucleation of pure water.
- When you precondition the aerosol at lower RH (10 and 1%) the aerosol nucleate ice close to the 'Koop line'.

I would like to better understand these findings because I am not sure I fully understand them. The figure below is taken from Lienhardt et al. (2015) for alpha-pinene aerosol. The yellow area is where the aerosol is in a 'glassy' state.

[Figure]

Preconditioning at 40% RH at room temperature should mean the aerosol are not in a glassy state (they are above the yellow area in the Lienhardt plot) and take on water as we expect according to Koehler theory. However, if the aerosol are then cooled to low temperature in the CFDC they should enter a glassy state with liquid water 'trapped' due to low diffusivity. On the other hand, preconditioning at low RH means that the aerosol will start in (or very close to) the glassy state at room temperature, and cooling further will lead to aerosol particles that have low water content in a glassy state.

Let us now consider what would happen to these particles in the CFDC. Both wet and dry particles have water contents below the threshold for homogeneous nucleation at these temperatures so should not nucleate ice at the start of the scan. The 'Koop' line sits around 85 to 90% RH, and the maximum preconditioned RH is 40%, so ice should not be nucleated, initially.

My understanding is that you then increase the RH in the CFDC during RH-scans. As mentioned above your data show that the "dry particles" nucleate ice on the 'koop line' whereas the "wet particles" nucleate ice close to the homogeneous freezing line for pure water.

The question I have is why do we see this different behaviour?

How do the "wet particles" get all the way to the homogeneous freezing line for pure water without nucleating ice, whereas the "dry particles" nucleate ice sooner?

The theory above suggests that the "dry particles" should have lower diffusion coefficients and therefore would struggle to increase their water content as the RH increases in the CFDC – it is more difficult for water to diffused through the dry aerosol particles (according to the Lienhardt plot).

My feeling is that the difference in size between the "wet" and "dry" preconditioned particles could be very important. As part of this review I did some back of the envelope calculations (assuming dry density is 1500 kg/m3 and dry molecular weight is 200 g / mole). These calculations suggest that under equilibrium conditions the preconditioned sizes are around 173 nm; 361 nm; and 652 nm respectively. Could it be that the water is able to diffuse into the smaller particles and dilute them sufficiently (so they nucleate on the 'koop line'), but not the largest particle size?

According to Figure 4 I think you may be suggesting that the wet preconditioned SOA forms an outer shell and traps the water inside, whereas the dry preconditioned SOA does not – This would explain your data, and so the idea about particle size might not be needed, but how does this happen? How and why is the dry shell formed on the outside of the wet preconditioned particles?

If this can be made clear in the manuscript I am happy to recommend publication.

---

## Author Comment (AC1)

**Response to RC1**

**Review of "Ice nucleation on surrogates of boreal forest SOA particles: effect of water content and oxidative age"**

We thank the reviewer for their comments and helpful suggestions. Our answers are in blue below. The tracked changes version of the manuscript has been posted as an Author Comment.

I have general comments on the abstract below. Generally, I think the paper is well written and a nice study that adds to the body literature. I also have a comment on figure 2, which is fairly trivial.

1. Abstract is a little longer than it needs to be. I feel like the first paragraph could be removed to add a bit more focus.

   As suggested, the first paragraph has been removed.

2. Was the core-shell formation observed in the measurements – could this be said explicitly in the abstract?
   Unfortunately, the core-shell formation can not be directly observed from the ice nucleation measurements. However, our observations constitute indirect evidence for the core-shell formation. In order to clarify this, the abstract has been modified as follows:

   Modified paragraph:
   *"The measurements suggest that at low temperatures, water diffusion into dry SOA particles is slow enough to form a core-shell morphology. The liquid outer layer can equilibrate within the timescale of the experiment and freeze homogeneously. On SOA particles with higher water content, water diffuses faster into the particle, delaying equilibration at the particle surface and preventing the formation of a diluted shell, which can delay homogeneous freezing. We propose that the partial water vapour pressure to which the particles are exposed prior to an experiment can serve as an indicator of whether a core-shell structure is developing".*

3. You clearly show the results of preconditioning the SOA at different humidities in figure 2. This is a key figure for the paper and well-presented. For those not well versed with the CFDCs I think it is necessary to explain what the steps in the plots are (scans). There is also a typo in the legend, which refers to 4% RH instead of 40%.
   - The typo in the legend has been corrected.
   - Added to Fig. caption:
     *"The color code indicates activated fraction"*

- The following sentences have been added to explain further how the measurements were performed:

    *"The activated fraction is measured along RH scans at several temperatures."*

That said, upon detailed reading I am confused about how it works.

The observations seem clear:

4. preconditioning the aerosol at 40% RH allows the aerosol to take on more water at room temperature. When you then transfer the aerosol to the CFDC these aerosol particles nucleate ice close to the threshold for homogeneous nucleation of pure water.

   In the 40% RH case, we did not observe ice formation in the covered RH range up to 100% water saturation (according to Murphy & Koop, 2005).

5. When you precondition the aerosol at lower RH (10 and 1%) the aerosol nucleate ice close to the 'Koop line'. I would like to better understand these findings because I am not sure I fully understand them. The figure below is taken from Lienhardt et al. (2015) for alpha-pinene aerosol. The yellow area is where the aerosol is in a 'glassy' state.

[Figure]

Preconditioning at 40% RH at room temperature should mean the aerosol are not in a glassy state (they are above the yellow area in the Lienhardt plot) and take on water as we expect according to Koehler theory. However, if the aerosol are then cooled to low temperature in the CFDC they should enter a glassy state with liquid water 'trapped' due to low diffusivity.

On the other hand, preconditioning at low RH means that the aerosol will start in (or very close to) the glassy state at room temperature, and cooling further will lead to aerosol particles that have low water content in a glassy state.

Let us now consider what would happen to these particles in the CFDC. Both wet and dry particles have water contents below the threshold for homogeneous nucleation at these temperatures so should not nucleate ice at the start of the scan. The 'Koop' line sits around 85 to 90% RH, and the maximum preconditioned RH is 40%, so ice should not be nucleated, initially.

My understanding is that you then increase the RH in the CFDC during RH-scans. As mentioned above your data show that the "dry particles" nucleate ice on the 'koop line' whereas the "wet particles" nucleate ice close to the homogeneous freezing line for pure water.

The question I have is why do we see this different behaviour?

How do the "wet particles" get all the way to the homogeneous freezing line for pure water without nucleating ice, whereas the "dry particles" nucleate ice sooner?

The theory above suggests that the "dry particles" should have lower diffusion coefficients and therefore would struggle to increase their water content as the RH increases in the CFDC – it is more difficult for water to diffuse through the dry aerosol particles (according to the Lienhardt plot).

My feeling is that the difference in size between the "wet" and "dry" preconditioned particles could be very important. As part of this review I did some back of the envelope calculations (assuming dry density is 1500 kg/m3 and dry molecular weight is 200 g / mole). These calculations suggest that under equilibrium conditions the preconditioned sizes are around 173 nm; 361 nm; and 652 nm respectively. Could it be that the water is able to diffuse into the smaller particles and dilute them sufficiently (so they nucleate on the 'koop line'), but not the largest particle size?

According to Figure 4 I think you may be suggesting that the wet preconditioned SOA forms an outer shell and traps the water inside, whereas the dry preconditioned SOA does not – This would explain your data, and so the idea about particle size might not be needed, but how does this happen? How and why is the dry shell formed on the outside of the wet preconditioned particles?

We think that the difference in ice nucleation between the "wet" and "dry" preconditioned particles cannot be attributed to a size difference of the particles, as the particles were

size-selected. According to the parametrization by Varutbangkul et al. (2006), the growth factors ($d_{wet}/d_{dry}$) for α-pinene SOA at 1%, 10% and 40% RH are 1.00000, 1.00008, and 1.007176 respectively, indicating that the preconditioned particle sizes barely differ from the dry size.

Regarding the suggested mechanism, our hypothesis is that the dry preconditioned SOA (Fig 1.a, below) forms an outer shell as a result of water uptake and slow diffusion into the particle. Even if the amount of uptaken water is small, the slow diffusion of the water into the particle (that takes place at low T inside SPIN) enables the formation of a liquid layer in equilibrium with the CFDC conditions. This outer liquid layer in equilibrium with the surrounding RH allows homogeneous freezing at the conditions marked by the "Koop line".

In the case of a "wet" particle, the water diffusion (Fig. 1.b, below) is faster, removing water from the surface.Assuming a comparable amount of water is taken up by the "wet" and "dry" particles during their 10 s residence time in the CFDC, the "wet" particle can not reach equilibrium and ends up more concentrated, too concentrated to freeze homogeneously. Equilibration with the surrounding humidity conditions would take more time. As equilibration time scales can be on the order of minutes and longer (Price et al., 2015), we think what we observed could also be happening in high updrafts in the atmosphere.

[Figure]

Fig 1. Suggested freezing mechanism for SOA particles with different water content.

If this can be made clear in the manuscript I am happy to recommend publication.
The discussion in section 3.1. has been modified:

*"When dry SOA particles (Fig. 4a) take up water at low temperature inside SPIN, water diffusion towards the particle core is slow enough to enable the formation of a liquid layer in*

*equilibrium with the CFDC conditions. This outer liquid layer in equilibrium with the surrounding RH freezes at the homogeneous freezing conditions (Koop et al. 2000). This mechanism is similar to the one suggested by Fowler et al. (2020).*

*For SOA particles with higher water content (Fig. 4b), water diffuses faster into the particle, removing water from the surface. Assuming a comparable amount of water taken up by the wet- and dry- conditioned SOA during their 10 s residence time in SPIN, wet-conditioned particles can not reach equilibrium, resulting in a more concentrated state, too concentrated to freeze homogeneously. Longer residence times could restore the homogeneous freezing of the particles by the equilibration with the surrounding RH. However, as equilibration time scales can be on the order of minutes and longer (Price et al., 2015), we believe that the observed inhibition of homogeneous freezing could also happen in fast updrafts in the atmosphere."*

---

## Author Comment (AC2)

**Review of "Ice nucleation on surrogates of boreal forest SOA particles: effect of water content and oxidative age" by Piedehierro et al.**

We thank the reviewer for their comments and helpful suggestions. Our answers are in blue below. The tracked changes version of the manuscript has been posted as an Author Comment.

**General comment:**

SOA particles have been shown to be important for the climate system, and therefore, if they are able to facilitate ice formation is of high importance. The present study reports the ice nucleation (IN) abilities of 100 nm SOA particles at temperatures between 210 and 240K using the SPIN. The role of oxidative age and water content on the IN abilities of the SOA particles was evaluated. The authors found that SOA particles are inefficient INPs and that the oxidative age has little effect on their IN abilities. On the other hand, the "water content" was reported to be a key driver in determining if SOA particles can nucleate ice particles via homogeneous freezing. The manuscript is well written, the experiments were carefully designed, and it nicely fits with the ACP scope. The manuscript can be accepted after the following points are considered.

**Major comment:**

- The authors argued that the "water content" was of high importance when determining if SOA particles can facilitate ice formation via homogeneous freezing or not. However, the "water content" was not measured/reported as it was simply inferred from the RH (>1%, >10% and 40%). It is of high importance if the authors can quantify the water content on the SOA particles at the three RHs.

  The SOA particles water content can be estimated from the moles of water and SOA ($M_{water}$, $M_{SOA}$):

  $$\text{Mole fraction water } [\%] = \frac{M_{water}}{M_{water} + M_{SOA}} \cdot 100 \qquad (1)$$

  where $M_{water}$, $M_{SOA}$ are calculated:

  $$M_{water} = \frac{4}{3}\pi r_0^3 \left(GF^3 - 1\right) \cdot \frac{1.0 \ g \cdot cm^{-3}}{18.01528 \ g \cdot mol^{-1}} \qquad (2)$$

  $$M_{SOA} = \frac{4}{3}\pi r_0^3 \cdot \frac{1.5 \ g \cdot cm^{-3}}{200 \ g \cdot mol^{-1}} \qquad (3)$$

for $r_0$ = 50 nm , using the hygroscopic growth factors (GF) for α-pinene SOA (Varutbangkul et al., 2006) and assuming an average molecular weight of 200 g mol $^{-1}$ and density of 1.5 g cm $^{-3}$. Results are summarized in Tab.1. The estimated mole fractions of water for the 1%, 10%, 40% RHw are 8.9 e-5% , 0.17%, 14% respectively.

**Table 1.** α-pinene SOA growth factors and water content (moles and mole fraction) at different equilibrium RHw.

|  | Growth factor (Varutbangkul et al., 2006) | Moles of water | Mole fraction of water (%) |
|---|---|---|---|
| **1% RHw** | 1.000000 | 3.5e-24 | 8.9e-5 |
| **10% RHw** | 1.000076 | 6.6e-21 | 1.7e-1 |
| **40% RHw** | 1.007176 | 6.3e-19 | 1.4e+1 |

To highlight the difference in particle water content at different RHs, the RHw=10% and 40% cases are now included.

*"The hygroscopic growth factors of α-pinene SOA at 10%and 40% RHw are 1.00008, and 1.00718 (Varutbankul et al., 2006). Assuming an average molecular weight of 200 g mol $^{-1}$ and a density of 1.5 g cm^-3 for the SOA particles,the corresponding water contents are 0.17% and 14% mole fractions, respectively."*

- The role of particle size on the ice nucleating abilities of SOA was not discussed at all. Ignatius et al. (2016) showed that there is particle size dependence on the SOA IN abilities. Why were 100 nm SOA particles selected for the present study? Also, the authors mentioned that the 100 nm SOA particles were quasi-monodisperse; however, particle size distributions were not provided.

  Ignatius et al. (2016) used polydisperse particles for their measurements. They note that: "We investigated SOA particles with mean diameters from 120 to 800 nm, and no dependency was observed between the particle size and the frozen fraction/freezing onset".

  For the current study, the focus was on the effects of particle composition. Monodisperse, 100 nm particles were used to exclude size effects. The choice of 100 nm as selected particle diameter was influenced by the size distributions produced in the PAM and ASC (see Fig.1). The median particle size of the SOA produced in PAM was 40 - 50 nm. 100 nm was the largest size at which a sufficiently high number of particles could be selected for all experiments. Because 100 nm is at the upper end

of the size distribution from PAM, the contribution of 151 nm (double charged) and 195 nm (triple charged) was negligible. In the ASC experiments, the median particle diameter was 90 - 110 nm and total concentrations were lower than for particle generation with PAM. Selecting 100 nm, close to the maxima of the size distribution ensured a sufficient number of aerosol particles for the experiments.

[Figure]

Fig 1: Typical size distribution during a PAM experiment (top) and an ASC experiment (bottom).

**Minor comments:**

1. In the introduction it is neither mentioned nor discussed why is it important to study SOA particles and what are the atmospheric implications if they act as INPs?

   We now mention the importance and implication in the first paragraph of the introduction:
   *" SOA acting as INPs affect precipitation formation, cloud cover and the cloud albedo. A better understanding of ice nucleation on SOA is of special relevance for predicting cloud properties in areas with boreal forest, where biogenic SOA can form in high concentrations."*

2. Recent and important studies relevant to SOA and ice nucleation such as Knopf et al. (2018); Wolf et al. (2020); Paramonov et al. (2020); Mahilang et al. (2021); Kilchhofer et al. (2021) and Bertozzi et al. (2021) are not discussed.

References to most of these relevant studies have been added to the text.

3. Along the text the authors mentioned that three different RH conditions were evaluated; however, it is unclear if the second RH was "<10%" or "=10%".

=10% is correct for the second measurement condition. Because ice nucleation results are similar for 10% and <1% RHw preconditioned particles, we refer to those cases as "dry" preconditioned or preconditioned at "low" RHw, corresponding to ≤10% RHw when grouped together.

Middle column label in figure 2 has been corrected to =10%.

4. Add Cziczo et al. (2013) in Line 5 page 2 in addition to DeMott et al. (2003).

The reference has been added.

5. Add Knopt et al. (2018) and Mahilang et al. (2021) after "ice nucleation mechanisms" in Line 15 page 2.

We added the Knopf et al., 2018 reference.

6. What were the typical OH and O3 concentrations during the SOA generation?

For PAM experiments, the $O_3$ concentration was 6.5 - 7 ppm at the inlet of PAM and the OH exposure was calculated with the model from Peng et al. (2015) to be 1-2 e12 molec cm-3. This can be converted to an average OH concentration of 5 - 10 e8 molec cm-3 inside the OFR. Note that this is not the true OH concentration. For that, detailed model calculations would be needed which were beyond the scope of this study.

The following sentences were added:

"Typical O3 concentration was 6.5 - 7 ppm at the inlet of PAM. The average OH concentration inside the PAM reactor was $5-10×10^8$ molec $cm^{-3}$ (estimated from OH exposure, Peng et al. (2015))."

7. The author mentioned that "low levels of background SOA (< 1 μgm−3)" were obtained between experiments. However, I am wondering how stable were the background particle concentration and particle size distributions?

There are two sources for background particles during PAM experiments: 1) contaminants in the operation gases which are oxidised and form particulate matter and 2) residuals inside the PAM reactor from previous experiments. These are "sticky" compounds which condensed on the inside of the reactor tube and may be volatilised in the next experiment due to changed conditions (e.g. higher OH concentrations). (1) was not an issue in these experiments as ultra high purity $N_2$ (from the headspace over liquid N2) and 5.0 $O_2$ were used as carrier/reagent gases. After 8 h of photochemical cleaning (i.e. no VOC introduced into PAM and OH lamp on experiment settings), no particles could be detected with an SMPS in a background test experiment. The contamination due to (2) was minimised by flushing the PAM with purified compressed air during idle times (i.e. between two experiments). Still a small amount of reactive vapours must have been deposited in the reactor tube. When the oxidants were introduced for the next experiment, SOA production was observed. However, the particles were very small (<20nm) and the amount of SOA quickly decreased. We waited until the 1μg $m^{-3}$ threshold was reached before continuing with the experiment preparations to make sure that no bias was introduced, e.g. for the determination of the OH exposure. We chose 1 μg $m^{-3}$ as this was 1% or less of the SOA mass produced during the PAM experiments. If there was a delay in experiment preparation, we could see this background decrease even further, even below the detection limit of the SMPS. Therefore, we are convinced that these background particles do not influence the SOA formation during the SPIN measurements.

8. Lines 23-26 page 6. Please clearly state that these results are from Schill and Tolbert (2012). As written it is not very obvious.

   The reference has been added again to make obvious the attribution of the statement.

9. "Humidity conditions" and "humidity needed" sound a bit awkward.

   Changed to RH- conditions.

10. Lines 17-19 page 9. How about particle size?

    Further research could include different particle sizes, to verify that the frozen fraction scales with surface area. Comment added to the text.

11. "This is the first study reporting the IN ability of SOA from real pine emissions representative of boreal forest environments." How about Paramonov et al. (2020)?

    The statement has been modified. Paramonov et al. (2020) reported INP concentrations in a boreal forest environment, but the nature of INP was not

identified. They investigated the INP concentration in the condensation/immersion freezing mode (at -31°C and 105% RHw). The data of our study shows that SOA from pine emissions do not contribute to INP at these conditions. Our study constitutes the first one in measuring the IN ability of pure SOA from real pine emissions in laboratory controlled conditions.

*"This is the first study measuring the IN ability of pure SOA from real pine emissions representative of boreal forest environments in laboratory controlled conditions."*

**Technical comments:**

1. I am not sure if "soots" is appropriate.

   Changed to soot.

2. Add more details about "purified air".

   The air purification system is custom built, using Wilkerson parts. The air is compressed and dried using a dryer. It is then filtered through particle filter (HEPA), active charcoal filter, and potassium permanganate filter.

   Details about the source of purified air have been added:

   *"...purified air (custom-built system, dry compressed air followed by particle, active charcoal and potassium permanganate filters)."*

3. "silica gel dryer". What was the length of the dryer, how often was it dried, and what was the RH at the end of the dryer?

   The silica gel dryer was 50 cm long. The silica gel was not dried during the measurement period, as it was not used for long periods of time and thus did not have time to get close to saturation point. THe RH after the drier was below 15%, well below the deliquescence point. Details have been added to the text.

4. "diffusion sources". What does it mean?

   There was a miscommunication about the details of the chamber operation. For the experiments presented here the liquid precursors (a-pinene or pine needle oil) were injected into a carrier gas flow with a microliter syringe. We corrected this part in the manuscript.

*"For experiments #13 and #15, α-pinene or pine needle oil were injected with a microliter syringe directly into a carrier gas flow. The injection times were adjusted to reach the desired 5 - 50 ppb in the ASC."*

A diffusion source (which was used in another set of ASC experiments which were not part of this study) consists of a small vial containing the liquid precursor with a capillary opening (i.D. e.g. 0.1 mm) placed in a larger container. There is a constant flow of purified compressed air through this container. Depending on the capillary size, room temperature, and air flow, a stable concentration of the precursor will be reached.

5. Add the model and brand of the used OPC.

   The OPC is part of the  SPIN setup sold by DMT. DMT does not provide an individual model or brand for it. For more details we now refer to Garimella et al. (2016).

6. Figure 3 caption. "AS data from Welti et al. (2020)". It seems to be out of place.

   The ammonium sulfate data used as reference and marked with crosses was taken from Welti et al. (2020).

7. Figure 5 caption. "viscosity 4e5 to 6e6". Fix it.

    It has been changed.